# Diversity and Abundance of Microbial Communities in UASB Reactors during Methane Production from Hydrolyzed Wheat Straw and Lucerne

**DOI:** 10.3390/microorganisms8091394

**Published:** 2020-09-11

**Authors:** Tong Liu, Anna Schnürer, Johanna Björkmalm, Karin Willquist, Emma Kreuger

**Affiliations:** 1Department of Molecular Science, Swedish University of Agricultural Science, Uppsala BioCenter, 750 07 Uppsala, Sweden; anna.schnurer@slu.se; 2RISE, Forskningsbyn Ideon Scheelevägen 27, 223 70 Lund, Sweden; johanna.bjorkmalm@borasregionen.se (J.B.); karin.willquist@ri.se (K.W.); 3Division of Biotechnology, Department of Chemistry, Lund University, P.O. Box 118, 221 00 Lund, Sweden

**Keywords:** wheat straw hydrolysate, microbial community, next-generation amplicon sequencing, UASB, ADM1, process acidification, methane production, VFA

## Abstract

The use of straw for biofuel production is encouraged by the European Union. A previous study showed the feasibility of producing biomethane in upflow anaerobic sludge blanket (UASB) reactors using hydrolyzed, steam-pretreated wheat straw, before and after dark fermentation with *Caldicellulosiruptor saccharolyticus*, and lucerne. This study provides information on overall microbial community development in those UASB processes and changes related to acidification. The bacterial and archaeal community in granular samples was analyzed using high-throughput amplicon sequencing. Anaerobic digestion model no. 1 (ADM1) was used to predict the abundance of microbial functional groups. The sequencing results showed decreased richness and diversity in the microbial community, and decreased relative abundance of bacteria in relation to archaea, after process acidification. Canonical correspondence analysis showed significant negative correlations between the concentration of organic acids and three phyla, and positive correlations with seven phyla. Organic loading rate and total COD fed also showed significant correlations with microbial community structure, which changed over time. ADM1 predicted a decrease in acetate degraders after a decrease to pH ≤ 6.5. Acidification had a sustained effect on the microbial community and process performance.

## 1. Introduction

Wheat straw is one of the most abundant agricultural residues in the world [1] and the most abundant in the European Union (EU) [2]. Use of wheat straw for biofuel production does not compete with land use for food production, and is encouraged by the EU Renewable Energy Directive 2009/28/EC [3] and its amendment EU 2015/1513 [4]. Wheat straw has been evaluated in different processes for microbial conversion to various energy carriers such as ethanol, hydrogen, methane, and combinations of these (e.g., [5,6,7,8,9,10,11,12,13,14,15,16]). Among these energy carriers, methane is highly interesting, as it can be used as a vehicle fuel (after upgrading the biogas) and for the production of electricity and other fuels and chemicals [17,18]. Moreover, the substrate conversion degree of complex substrates (containing many different and intertwined compounds) has been shown to be higher for methane production through anaerobic digestion (AD), alone or in combination with other biofuel processes, than in the production of either ethanol or hydrogen [13]. The environmental impact of the entire methane production and utilization chain is also relatively low [13,17].

Anaerobic digestion of organic material is a complex microbiological process requiring the combined activity of several groups of microorganisms with differing metabolic capacities (as reviewed in [19]). The degradation process can be divided into four main steps: hydrolysis, acidogenesis, acetogenesis, and methanogenesis. These are performed by the combined action of three physiological groups of microorganisms: hydrolytic–acidogenic bacteria, syntrophic acetogenic bacteria, and methanogenic archaea. To obtain a stable and efficient biogas process, it is important to meet the growth requirements of all microorganisms involved and here, the substrate is one critical parameter. The composition and dynamics of the involved microbial community are also strongly related to operating parameters such as process temperature, retention time, and organic loading rate [19]. Many studies have investigated correlations between microbial composition and digester performance in terms of microbial community structure and have shown that the microbiome has a high functional redundancy and robustness and a remarkable ability to adapt to various operational conditions, of importance for process stability [19].

In AD of lignocellulosic particulate materials, such as straw, hydrolysis is in general the rate-limiting step [20,21,22]. Although a wide variety of organic compounds present in lignocellulosic substrates can be degraded in AD, the presence of lignin and other recalcitrant compounds limits the rate and degree of hydrolysis of degradable compounds [23]. One proven route for increasing the substrate conversion rate and degree of wheat straw in AD is the use of physiochemical pretreatment, e.g., acid-catalyzed steam pretreatment [24] or other hydrothermal treatment [9], making the material more accessible for microbial degradation. Subsequent enzymatic hydrolysis, with added enzymes, to liquefy the abundant cellulose and hemicellulose, does facilitate the use of a high-rate AD reactor, such as an upflow anaerobic sludge bed (UASB) reactor [5,9,11,24,25]. Willquist et al. [14] used the anaerobic digestion model no. 1 [26] to predict the positive influence of including dark fermentation as a pre-step, with microbes that convert some of the sugars to hydrogen and acetate, which can have additional beneficial effects on the AD process. Degradation of sugars in the acidogenic/acetogenic steps results in the production of various organic acids, hydrogen, and carbon dioxide [19]. If the rate of product formation is higher than the preceding conversions, there is a risk of accumulation of acids, such as propionic and butyric acid in the process, which could result in process acidification [26]. In the current paper the term “acidification” refers to a decrease of pH of the AD reactor liquid to below pH 6.2 and “pre-acidification” refers to formation of acids prior to the main AD reactor. Oxidation of e.g., propionate and butyrate are only thermodynamically favorable (∆G’ < 0) at P_H2_ < 10^−5^ bar [26]. A low sugar to acetate ratio in the feed was proposed to decrease H_2_ production during the AD process, and consequently, lower the partial hydrogen pressure (P_H2_), which could improve the anaerobic oxidation of different organic acids [14]. In addition, having only sugars in the feed, with no volatile fatty acids (VFAs), has been shown in several studies to result in fragile granules with poor settling in UASB reactors (as reviewed in [27]). However, it has been demonstrated that some sugars are still needed for good granulation in UASB reactors (for thermophilic AD), when the feed is composed of only VFAs [28]. In combination, these findings suggest that partial pre-acidification (which can be done with dark fermentation or by other means), rather than no pre-acidification or complete pre-acidification for AD, could have beneficial effects on the process in an UASB reactor.

Increased share of acetic acid in relation to sugar in the feed could reduce the risk of acidification and improve granular structure, as described. However, it might also lead to a higher concentration of acetic acid in the reactor which, if the alkalinity is not sufficient, can lead to a decrease in the pH and an increase in the undissociated form of acetic acid and other VFAs. The undissociated forms of several VFAs, including acetic, propionic, and butyric acids, have been shown to be inhibitory to a wide variety of bacteria and methanogens, to varying degrees, while the dissociated forms are generally considerably less inhibitory [29,30,31,32,33,34,35,36,37]. Operating strategies leading to reduced risks of acid accumulation and pH drop are thus desirable.

In a previous study, in which UASB reactors for AD were operated on hydrolyzed, steam-pretreated wheat straw and lucerne, with and without a thermophilic dark fermentation pre-step for the hydrolyzed wheat straw, these reactors showed differences, specifically related to acidification events [5]. During dark fermentation with *Caldicellulosiruptor saccharolyticus*, acetic acid was formed in parallel with hydrogen and buffering agents were added to neutralize the acid during fermentation [5]. Therefore, the alkalinity of this effluent was higher than in liquid from material not passing through dark fermentation, despite similar pH when fed to AD. Consequently, the UASB reactors fed the substrate without a dark fermentation pre-step acidified to a pH below 6 with a maximum organic loading rate (OLR), expressed as a chemical oxygen demand (COD) of 7 g/L/d [5]. However, processes operated with a substrate where a dark fermentation pre-step had been applied could be operated at a higher OLR without acidification.

The aim of the present study was to obtain information on microbial communities in the above mentioned UASB reactors and to reveal knowledge on the response to substrate composition and increasing organic load as well as acidification events. Knowledge on the microbial community and its responses can provide valuable information on process performance, explain observed variations, and aid in long-term process management [38,39]. Many studies have investigated the microbial community composition in CSTR processes with the aim to understand its response to various operating parameters and performance variables (as reviewed in [19]). However, less knowledge about microbial communities in UASB reactors is available and studies related to operation with high strength substrates, as in the present study, are scarce [35,36,40]. In addition, only few previous studies have investigated the effects of increasing loads using high COD substrates [36,40]. These previous studies show good possibilities to treat such substrates in UASB systems but also risk VFA accumulation, acidification, and reduced methane production efficiency. Microbial communities in these previously studied processes, operating with hydrolyzed brewer’s spent grain or sugar refinery wastewater, illustrated good capability in response to the increasing loads, with changes in both composition and distribution of both fermentative acidogenic bacteria, particularly members within phylum Bacteroidetes, and methanogens [36,40].

To deepen the knowledge related to microbial responses in UASB reactors, archaeal and bacterial communities were in the present study analyzed by high-throughput amplicon sequencing of 16S rRNA genes by the Illumina MiSeq platform. Quantitative polymerase chain reaction (qPCR) was additionally conducted to quantify key groups of methanogens. The results were related to operating variables such as OLR, pH, organic acids, undissociated propionic acid, and alkalinity as well as to the total COD fed and compared with previous studies on UASB processes. Standard expressions and parameter values from the ADM1 [26] were used to evaluate how well the observed changes with the microbial community could be illustrated through standard kinetic growth and inhibition expressions in the model.

## 2. Materials and Methods

### 2.1. Operation of UASB Reactors

Reactor set-up (Appendix A) and overall performance and monitoring of four UASB reactors (1A, 1B, 2A, 2B) for part of the operating period were as described previously by Byrne et al. [5]. For interpretation of the microbiological results, a short summary of these results is given here. In addition, new results are presented for a longer operating period of reactors 2A and 2B, including adjustments at pH drops. The results in [5] are based on operating days 58–67 for 2A and for 2B. The reactors (0.86 L active volume) were initially inoculated to half based on wet weight (ww), with granules from an internal circulation anaerobic digestion reactor (in Falkenberg, Sweden) treating wastewater from beer and soft drink production and operated by Vatten & Miljö i Väst AB (Falkenberg, Sweden). The other half of the granules (ww) originated from the same plant, but had been used for a few initial experiments employing the same conditions as in [5], except for temporary additions of different levels of trace metals and addition of a 30–100% more concentrated substrate, with no addition of lucerne [unpublished].

For the experiment reported here, the operating temperature was 37 °C except in the first 5 days and during period 3 (Table 1). Reactors 2A and 2B were operated with combined hydrolysate-1 (CH-1), composed of wheat straw hydrolysate (WSH) and enzymatically hydrolyzed, steam-pretreated ensiled lucerne (LH), water, NaOH, and micronutrient supplements. The concentrations of COD and organic acids (see definition in 2.3) in CH-1 were: COD 36 g/L, acetic acid 1.85 g/L, lactic acid 0.12 g/L, succinic acid 0.48 g/L, and valeric acid 0.33 g/L. Reactors 1A and 1B were initially operated with the same substrate (CH-1), but switched to combined hydrolysate dark fermentation effluent (CH-DFE) at day 58. CH-DFE consisted of LH, effluent from dark fermentation of WSH, NaOH, micronutrient supplements, and water. The concentrations of COD and organic acids in CH-DFE were: COD 34 g/L, acetic acid 4.71 g/L, lactic acid 0.57 g/L, succinic acid 0.36 g/L, and valeric acid 0.33 g/L. Methods used for pretreatment, hydrolysis, and dark fermentation (with *Caldicellulosiruptor saccharolyticus*) are described by Byrne et al. [5]. Details of substrate composition and concentrations of nutrients in effluents are given in [5] for the entire operating period of reactors 1A and 1B, and for reactors 2A and 2B, up until acidification of 2B at day 71. Addition of alkaline agents during acidification and thereafter for 2A and 2B is given below. From day 161, a solution of copper and zinc was added to reactors 2A and 2B, at concentrations based on the analysis of effluent concentrations [5]. Zinc (as ZnCl_2_) and copper (as CuCl_2_), corresponding to a final concentration of 1 mg/L and 0.1 mg/L, respectively, per volume feed were added by adding 100-fold concentrated solution directly to the reactors 1–2 times per week.

The WSH and LH, both with pH 4.8, were collected in two separate 60 L barrels after production, stored at 6 °C for a maximum of three weeks, then aliquoted in 1 L batches and stored frozen (−20 °C). The dark fermentation effluent (DFE), pH 6.5, produced with a defined culture, was collected in glass bottles (with microfilter to the surrounding air) over 6 days at room temperature and then, stored at 6 °C, until all batches were collected. After centrifugation, it was aliquoted in 4.8 L buckets and stored frozen (−20 °C). WSH, LH, and DFE were thawed and stored at 6 °C for a maximum of 2 weeks prior to preparation of the final substrates. The final substrates CH-1 and CH-DFE were prepared in 3 L batches, one by one prior to use, and stored at 6 °C for a maximum of 2 weeks (normally less than 1 week) prior to filling the substrate vessels, plus an additional maximum of 1 week (less at higher OLR) at 4–6 °C in the substrate vessels.

Substrate was fed to (and effluent was removed from) each reactor for 6–32 min in 18 h of the day (no feeding at 03.00, 07.00, 11.00, 15.00, 19.00, and 23.00 h). Samples were taken 30–90 min after one feeding for analysis of pH, alkalinity, and concentrations of COD, organic acids (formic, acetic, propionic, butyric, valeric, lactic, and succinic acids), and other metabolites not reported here, as described in [5]. The stability of organic acid concentrations and other metabolites in the substrate while stored in cooled feed vessels was analyzed once (day 147), 4 days after feed vessels were filled. The influence of sampling time after feeding on reactor liquid composition was analyzed by multiple sampling over time after feeding (30, 60, and 90 min) on days 42, 49, and 215. The liquid level in the reactors was manually checked and replenished by the addition of 10–100 mL of effluent liquid when required. As described in [5], a different method for effluent removal than the pumping applied would have been preferred, since a temporary decrease in the liquid level below the effluent recirculation can disrupt recirculation.

The experimental period was in total 217 days, divided into four periods (P1–P4) based on OLR and substrate type (Table 1 and Figure 1). The accumulated mass of COD fed per liter reactor is shown in Appendix A. During P3, none of the reactors received any feeding, for personal non-scientific-based reasons of the operator, and the temperature was initially reduced to room temperature, around 25 °C, and recirculation was stopped. On day 117, after 39 days without recirculation, the granular bed height (mean of the highest and lowest point around the circumference) was determined and volume was estimated by comparison with an empty reactor (maximum and minimum volumes are denoted ±). The temperature was gradually increased back to 37 °C at the end of this period. During operation, feeding was in general decreased or stopped when the pH decreased below 6.8 and the OLR was above COD 2 g/L/d. However, on a few occasions, feeding continued despite a recorded pH 6.6–6.7, for reactor 1B on day 57; 2A on day 74; and 2B on days 140–147. The reason for this was that the pH limit for decrease or stopping of organic loading was not set sharp prior the experimental start.

#### 2.1.1. Treatment of Reactor 2B at Acidification During P2

On day 71–75, additional measures were taken to increase the pH above 6.8 in reactor 2B, due to a recorded pH of 4.7. Over the course of 2 h on day 71, in total 0.776 g NaOH and 1 g/L_reactor_ NaHCO_3_ in 80 mL H_2_O and 260 mL effluent from 2A (pH 6.9) were added stepwise to 2B and reactor liquid was removed. On day 72, 1 g/L_reactor_ NaHCO_3_ in 30 mL H_2_O was first added to 2B, and then, 420 mL reactor liquid was exchanged stepwise with 420 mL effluent from 2B collected on days 49–57 (pH 7.4). Finally, another 1 g/L_reactor_ NaHCO_3_ in 10 mL H_2_O was added on day 72, while on day 75, another 360 mL reactor liquid from 2B was exchanged with effluent from 2B from days 49–57.

#### 2.1.2. Unusual Events during P4

A stop in recirculation was noted for 1A on day 138. The length of the period without recirculation is unknown (maximum 4 days), but it was started again 3–4 h prior to sampling. The recirculation pump for 1B was off (by mistake) on days 205–206, so feeding to that reactor was stopped for 10 h. It was sampled for COD concentration after 10 h with recirculation and without feeding. The pH of substrate CH-1 was adjusted with NaOH solution (194 g/L) to 7.13 (instead of 7.0) on days 143–180 and to pH 8.3 on days 161–173.

### 2.2. Sampling of Microbial Granules

Microbial biomass from all reactors was sampled on days 58, 78, 126, 153, and 204, and from 1B also on day 218 (Figure 1). On each sampling occasion, 10–20 mL was drained from the bottoms of the reactors into beakers, by spontaneous flow, after opening the bottom tubing. From the sample, 2 × 4 mL granules were drained with a spoon towards the side of the beaker and transferred to tubes. Excess liquid and granules were returned to the bottom of the reactors with a syringe.

### 2.3. Analytical Methods

The methods used for analysis of COD, alkalinity concentrations, pH, methane volume, and concentrations of metabolites are described in [5]. In short, COD was determined with LCK114 test kits (HACH, Loveland, CO, USA), methane production with AMPTS (Bioprocess Control AB, Lund, Sweden), pH with a pH meter (off-line), total and partial alkalinity with titration, and the concentrations of acids and other metabolites by HPLC (Jasco Co., Tokyo, Japan) using an Aminex HPX-87H column (Bio-Rad, Hercules, CA, USA) and refractive index detector (Shimadzu Co., Kyoto, Japan). Here, “VFAs” refers to formic acid, acetic acid, propionic acid, butyric acid, and valeric acid, while “organic acids” refer to the VFAs, lactic acid, and succinic acid. The method for the detection of organic acids and other metabolites is described in [5]. The concentration of undissociated propionic acid at 37 °C was calculated based on total propionic acid and a pKa value of 4.88 (the average of pKa at 35 and 40 °C, based on [41]).

### 2.4. DNA Extraction and Microbial Community Analysis

Triplicate 0.2 mg portions of wet granule samples were used to extract total genomic DNA following the protocol of FastDNA Spin Kit for Soil, MP Biomedicals (Valiant Co., Ltd., Yantai, Shandong, China), with an additional wash step to remove humic acid. The universal primer sets 515’F(GTGBCAGCMGCCGCGGTAA) and 805R(GACTACHVGGGTATCTAATCC) were used to amplify the 16S rRNA genes of both archaea and bacteria [42], and specific primer sets 516F(TGYCAGCCGCCGCGGTAAHACCVGC) and 915R(GTGCTCCCCCGCCAATTCCT) were used for amplification of archaeal 16S rRNA genes [43]. The preparation steps in building the amplicon libraries for next-generation amplicon sequencing are described in a previous study, in brief, including two rounds PCR, gel run for size confirmation, and PCR products purification [44]. Sequencing was performed at SciLifeLab in Uppsala, Sweden, using the Illumina MiSeq platform. The raw DNA sequencing data obtained were submitted to the National Center for Biotechnology Information database (NCBI) under accession numbers—bacteria: from SRR7767592 to SRR7767633; archaea: from SRR7782695 to SRR7782766—were analyzed through the open source bioinformatics pipeline DADA2 (version 1.8) [45]. In detail, the DADA2 pipeline was performed in R (v 3.4.0; https://www.r-project.org/) using module ‘cutadapt’, packages ‘DADA2’ and ‘phyloseq’. The primers and index were first trimmed by ‘cutadapt’ and any sequencing containing N base was discarded. Based on the plotted quality profile, the forward and reverse sequences were cut to lengths 250 and 150 bp, respectively. Sequences not passing the quality threshold (maxEE = 2, truncQ = 11) were discarded. Chimeric sequences were then discarded from the paired-end and dereplicated sequences. These non-chimeric sequences were used for inferring bacteria and archaea taxa against the SILVA ribosomal RNA gene database (v128). Finally, unique taxa were inferred by DADA2 and used to build tables of amplicon sequence variants (ASVs) [45]. Microbial community richness and diversity were evaluated using Hill diversity index ^0^D and ^1^D, respectively [46,47], in the R package ‘hillR’ (https://CRAN.R-project.org/package=hillR). Permutational ANOVA (PERMANOVA) was performed to select significant process operating parameters affecting microbial communities, and these parameters were further used for beta diversity analysis (canonical correspondence analysis, CCA), using R package ‘vegan’ (https://CRAN.R-project.org/package=vegan). Methanobacteriales were quantified by qPCR, using the primer set MBT. The program used for qPCR and the data analysis protocol was as described previously [48].

### 2.5. Modeling

ADM1 [26] was used to evaluate the effect of putative pH changes on the microbial population. In ADM1, there are several dynamic state variables representing different groups of microorganisms, including acidogens (Xsu, Xaa, and Xfa), acetogens (Xc4 and Xpro), hydrogen degraders (Xh2), and acetate degraders (Xac). The aim was to determine whether the ADM1 model could describe the changes in microbial population following overloading and subsequent acidification. The model was implemented in MATLAB R2017a (Mathworks, USA) according to the procedure in [49]. The characteristics of the substrates (CH-I and CH-DFE) used in the experiments were added to the model, i.e., sugar and organic acids content (Table 3 in [5]) and nitrogen and protein (Table S1 in [5]). The inhibition expression of VFA inhibition was summarized as a lower pH inhibition expression, as is the standard setting of the ADM1 [26]. The model parameters were taken from previous publications [26,49]. An OLR of 6 g/L/d was applied in the model and pH was subsequently changed from 5.5 to 8.0 in 0.5 steps. For each pH change, the simulation was run until steady state was observed. No calibration of parameters or adjustment to upflow dynamics was performed. The pH sensitivity of the process is estimated using the lower limit pH inhibition equation according to [26]. When pH < pH_UL_ (pH_UL_ is the upper limit pH value), Equation (1) is used and when pH ≤ pH_UL_, Equation (2) is used.
(1)$$I\;=\;\exp\left[-3{\left(\frac{\mathrm{pH}-{\mathrm{pH}}_{\mathrm{UL}}}{{\mathrm{pH}}_{\mathrm{UL}}-{\mathrm{pH}}_{\mathrm{LL}}}\right)}^{2}\right]$$
I = 1(2)

The (pH_UL_:pH_LL_) parameter value from [26] was used in this study.

## 3. Results

### 3.1. UASB Process

The process performance of the UASB reactors during periods of stable operation is described in [5]. Below, more details on process variables over the entire operating period are given, to assist in interpretation of data on the microbial populations.

#### 3.1.1. Periods 1, 2, and 3

Towards the end of P1, when all four reactors were operated with the same substrate (CH-1) at increasing OLR (from COD 2.0 to 5.3 ± 0.2 g/L/d), the processes showed similar behavior with regard to pH, concentrations of VFAs, COD reduction, and partial alkalinity (Figure 2 and Figure 3, Appendix A). The organic acids were dominated by the VFAs acetic acid, propionic acid, and butyric acid (Figure 3). The variations in concentrations of these VFAs were small in absolute terms in samples taken 30, 60, or 90 min after a feeding pulse (Appendix A). Formic acid and lactic acid were under the detection limit in all reactors over the entire operating period. Succinic acid, glucuronic acid, and galacturonic acid were found occasionally (results not shown), with maximum concentration of 0.12 g/L apart from succinic acid in reactor 2B at day 71 (0.3 g/L).

At the start of P2, the substrate was changed for reactor 1A and 1B. In response to this change, the partial alkalinity in 1A and 1B reactors increased towards the end of the period, to CaCO_3_ 3.4 ±
0.2 (n=4) g/L (average of day 70 and 71), compared with CaCO_3_ 1.7 ± 0.0 g/L (n=2) in reactor 2A on day 70–71. The change was triggered by the higher alkalinity of the substrate CH-DFE compared with CH-1, caused by the addition of buffering agents during the dark fermentation process [5]. During P2, the pH in processes 1A and 1B also increased to 7.2, while in 2A, it remained around 6.8 (Figure 2). The concentrations of VFAs were low in all these three processes (Figure 3) at similar OLR (Figure 1 and Table 1). However, reactor 2B deviated from the other reactors, with increased concentrations of VFAs, reduced alkalinity and pH, and acidification to pH 4.7 (Figure 2 and Figure 3, Appendix A). The concentration of organic acids reached 6.7 g/L and the undissociated propionic acid concentration 870 mg/L on day 71 (Figure 3). Moreover, some of the granules in 2B disintegrated and microbial biomass was lost in the effluent (see below). By repeated addition of base and exchange of reactor liquid, the pH in reactor 2B was increased and stabilized above pH 6.8, with simultaneous decrease in the concentrations of organic acids to 2.3 g/L and undissociated propionic acid to 9.6 mg/L on day 77 (Figure 2 and Figure 3).

In the following starvation period (P3), the concentrations of all organic acids analyzed, except valeric acid (0.1 g/L), decreased to non-detectable levels in all reactors and the pH increased above 7. The volume of the granular beds of UASB reactors 1A, 1B, 2A, and 2B towards the end of P3 was: 218 ± 6, 206 ± 6, 267 ± 7, and 142 ± 10 cm^3^, respectively. Thus, the granular bed volume of reactor 2B was around 35%, 47%, and 31% lower than that of 1A, 1B, and 2A, respectively.

#### 3.1.2. Period 4

During the final operating period (P4), the organic loads started at a low level and were increased successively (Figure 1 and Table 1). On some occasions, acids accumulated in the different reactors (Figure 3). After an uncontrolled stop in recirculation for 1A, noticed on day 138, VFAs accumulated from 0.1 to 2.3 g/L, the HAc/PA ratio (ratio of acids to partial alkalinity as g acetic acid equivalents to g CaCO_3_) increased from below 0.1 to 0.7, and undissociated propionic acid concentration increased from below detection limit to 9.2 mg/L (Figure 3 and Appendix A). On day 138, the total propionic acid concentration in 1A was 1.4 g/L, which was similar to that in 2B (1.5 g/L) when the liquid of that reactor was acidified on day 71 (pH 4.7). However, since the pH remained close to neutral in 1A, the concentration of undissociated propionic acid was much lower in 1A on day 138 (9.2 mg/L) than in 2B on day 71 (870 mg/L). At the next measurement point in reactor 1A, i.e., day 151, the concentration of undissociated propionic acid had decreased to 1.3 mg/L and total propionic acid to 0.3 g/L. However, during a peak in OLR of COD 12.7 g/L/d on days 203–204 (Figure 1), the pH decreased to 6.7–6.8 and the level of undissociated propionic acid again increased (Figure 2 and Figure 3). The feeding to reactor 1A was then decreased and ultimately stopped on day 210.

At day 204–205, the recirculation pump in 1B was off and the pH at this time was 6.8. Feeding was stopped for 10 h and during this time, the pH increased again to above 7. Thereafter, the concentration of VFAs decreased from day 208 to 215, and the pH was neutral and stable until termination of the experiment on day 217. The average OLR was COD 8.5 ± 0.2 g/L/d for 1A on days 183–201 and for 1B on days 208–217.

When feeding was started in P4, some accumulation of acetic and propionic acid, with higher levels in 2B compared with 2A, was seen after only one week, with a volumetric OLR slightly above COD 2 g/L/d. The pH in reactor 2B also decreased to pH 6.6 only three days after the feeding was started. The organic load per unit microbial biomass in this reactor was higher than in the other reactors, due to the substantial loss of microbial biomass from 2B at acidification during P2. In order to increase the alkalinity in reactor 2A and 2B liquid, the pH of substrate CH-1 was increased slightly to pH 7.1 on day 143 and to pH 8.3 on day 161. The ratio of undissociated acids to total acids in the reactors was kept lower after this action. However, the accumulated acids were not completely degraded and pH dropped below 6 in both reactors on day 172 (pH 5.1 in 2A and 5.9 in 2B). On day 168 and day 173, the feeding to 2B and 2A, respectively, was finally stopped.

The concentration of organic acids in the substrate after 4 days in the feed storage vessels, since preparation, was analyzed on day 147. The concentration of valeric acid was found to be stable. The concentration of lactic acid and succinic acid changed by a maximum of 0.06 g/L and that of acetic acid by a maximum of 1%. The ethanol concentration increased markedly in CH-DFE, from 0 to 0.93 and 0.72 g/L in the feed vessels of 1A and 1B, respectively.

### 3.2. Microbial Communities

#### 3.2.1. Diversity Indices

Zero-order (^0^D) and first-order (^1^D) Hill diversity index were used to represent microbial community richness and diversity, respectively [46]. The value of ^0^D across microbial samples (generated using universal primer set 515’F and 805R) varied from 138 to 388, with the lowest values, among all reactor samples, for reactor 2B at day 126 (in P3) and the highest for 1A at day 57 (at the end of P1) (Appendix A). For archaea samples (generated using archaeal primer set 516F and 915R), ^0^D varied from 17 to 71, with the lowest values in the original inoculum and the highest in reactor 1A at day 204 (P4) (Appendix A). In general, the overall microbial community over time showed a trend for a decrease in both richness and diversity. This decrease was more significant in reactor 2B than in other reactors, in which ^0^D dropped from 238 to 185, and ^1^D index decreased from 63.1 to 22.9 (Appendix A). For archaea, a similar downward trend in ^1^D was seen over time in all reactors. However, ^0^D increased in 1A and 1B (from on average 26.7 at day 58 to 67 at day 204), while in 2A and 2B, it stayed at a similar level over time (36.3 ± 3.6 and 26.7 ± 2.8, respectively), with the lower value in 2B (*t*-test, *p* < 0.01) (Appendix A).

#### 3.2.2. Correlation of Microbial Community Structure with Process Variables

Canonical correspondence analysis was performed using ASVs generated by universal primer set 515’F and 805R to reveal possible general correlations between microbial community composition within the reactor samples and biogas process operating variables (Figure 4). The first two dimensions of the CCA together explained around 68.9% of variation in the relative abundance of ASVs. The variables organic acids, HAc (acetate content), and HPr (propionate content) had the most significant influence (*p* < 0.01) on the overall microbial community in the reactors, followed by undissociated HPr (*p* < 0.02) (arrow overlapped with HPr), OLR (*p* < 0.05), HPr_to_HAc (propionate to acetate ratio), and total COD fed (*p* < 0.1). Specifically, the relative abundance of the phyla Planctomycetes, Omnitrophica, and Deferribacteres showed strong positive correlations with total COD fed (Figure 4), with a significant increase in Planctomycetes on day 204 in reactors 1A and 1B, from 0 and 1.9% to 9.4 and 13.5%, respectively (Figure 5). The phyla Euryarchaeota, Lentisphaerae, Thermotogae, Tenericutes, WWE3, Aminicenantes, and RBG−1 (Zixibacteria) showed positive correlations with the concentration of total organic acids, while the phyla Synergistetes, Chloroflexi, and Proteobacteria showed the opposite (Figure 4). The phylum Gemmatimonadetes, BRC1, was also positively correlated with increasing OLR (Figure 4). However, the relative abundances of the phyla Gemmatimonadetes, Omnitrophica, Deferribacteres, Lentisphaerae, WWE3, Aminicenantes, and RBG−1 (Zixibacteria) were low, less than 1%, and thus, they are shown merged as ‘Minor phyla’ in Figure 5 (The data presented in Figure 5 are also visualized as an abundance heatmap in Appendix A, and on ASVs level in Appendix A).

After P1 (day 1–57), all four reactors developed similar microbial communities as in the original inocula. Within the bacterial community, the most dominant phylum on average (at sampling day 58) was Synergistetes (29.9 ± 7.4%), followed by Firmicutes (14.7 ± 2.7%), Proteobacteria (11.9 ± 2.9%), Bacteroidetes (7.9 ± 0.5%), and Spirochaetes (6.2 ± 2.3%). Within the archaeal community, Euryarchaeota was the only dominant phylum (15.5 ± 6.6%) (Figure 5).

In P2 (day 58–77), significant changes were seen in 2B after acidification day 71. After the substrate switch to CH-DFE in reactors 1A and 1B, Synergistetes increased in these reactors (from 19.2 to 38.9% and 34.3 to 36.9%, respectively). In contrast, the relative abundance of these phyla showed a decrease in reactors 2A and 2B (from 30.9 to 25.1% and 35.1 to 11.0%, respectively) (Figure 5). After the substrate switch to CH-DFE in 1A and 1B, no significant changes were seen in relative abundance of Firmicutes and Bacteroidetes in P2 (*t*-test, *p* > 0.1), while there was a slight decrease in Proteobacteria across all four reactor samples, from 11.9 ± 2.9% to 6.7 ± 2.6% (*t*-test, *p* < 0.1). A small change was also seen for the phylum Synergistetes, which increased slightly in reactors 1A and 1B, while the opposite was seen for reactors 2A and 2B. In P2, Euryarchaeota showed a decrease in relative abundance in 1A, but an increase in the other reactors, with a notable increase in 2B from 10.9 to 63.5% (Figure 5).

During P3, when the feeding was stopped (day 78–129), little effect was seen on the bacterial community. However, a gradual increase in relative abundance of Euryarchaeota was seen in 1A, 1B, and 2A, but not 2B where this phylum was already the most dominant since P2 (Figure 5). In P4 (day 131–217), more pronounced changes in the microbial community were seen in all four reactors, but with differences between the reactors. These changes varied in the microbial groups, corresponding with the different levels of OLR and organic acids. From sampling day 153 to 204, Firmicutes and Bacteroidetes increased dramatically. Firmicutes increased from 9 to 25.9% in 1A, and Bacteroidetes increased from 7.2 to 32.4% in 2A (Figure 5). These increases were attributed to enrichment of an unclassified genus of the family Eubacteriaceae (belonging to Firmicutes) and the genus *Alistipes* (belonging to Bacteroidetes) (Appendix A). Synergistetes (mainly represented by the genus *Aminivibrio*) decreased in all reactors compared with the previous period, except in reactor 1A (Appendix A). The relative abundance of Synergistetes was also higher in 1A and 1B (mean 32.1 ± 10.4%) than in 2A and 2B (mean 18.2 ± 9.8%) (*t*-test, *p* < 0.01) (Figure 5). Proteobacteria (mainly represented by the order Rhizobiales) also continued its decreasing trend in all four reactors during P4, with a mean relative reduction in abundance from the initial level of 11.9 ± 2.9% (P1) to less than 1% (Appendix A). Moreover, during P4 (since sampling day 204), the relative abundance of the phylum Planctomycetes (represented by an unidentified clone vadinHA49) increased from less than 1% to on average 11.8 ± 2.2% in reactors 1A and 1B. The phyla Tenericutes and Thermotogae appeared in 2A and 2B at the end stage, at an average relative abundance of 1.9 ± 0.2% and 2.8 ± 1.4%, respectively (Figure 5).

For archaea, Euryarchaeota was at a similar level at the end of P4 as it was in P3 in all reactors except 1A, in which the relative abundance drastically decreased from 24.8 to 1.1% (Figure 5). Euryarchaeota was represented by the genera *Methanobacterium* and *Methanosaeta*, with much higher relative abundance of *Methanobacterium* across all reactor samples (Appendix A). A decrease was seen in *Methanosaeta* in 2A and in 2B after acetate accumulation and pH drops below 6 (Appendix A). However, qPCR analysis showed the Methanobacteriales (represented by the genus *Methanobacterium*) in general stayed at a similar level over time in all four reactors (Appendix A).

Use of the archaeal primer set (516F and 915R) revealed similar results to the universal primer set (515’F and 805R), i.e., (1) dominance of Euryarchaeota, (2) higher relative abundance of *Methanobacterium* than *Methanosaeta*, and (3) lower relative abundance of *Methanosaeta* in 2B (Appendix A). The archaeal primer set also detected the phyla Bathyarchaeota and Woesearchaeota. Bathyarchaeota showed a slight increase in reactors 1A and 1B, from less than 1% at day 58 to on average 4% at day 204 (periods 2–4), while Woesearchaeota increased in 1B and 2A at day 204, also from less than 1% to on average 5.1% (Appendix A).

### 3.3. Modeling

Anaerobic digestion model no. 1 (ADM1) was used to predict acid concentrations and the abundance of functional groups of microorganisms and their response to pH at a given OLR of 6 g/L/d. The model predicted considerably higher concentrations of acetic acid at pH 6.8 than at pH 7.2. For CH-1, the acetic acid concentration was predicted to be 1–2 g/L higher, and for CH-DFE 2–3 g/L higher, at pH 6.8 compared with pH 7.2 (Appendix A).

Acetate degraders (including acetoclastic methanogens) were predicted to be more abundant than hydrogen utilizers (including hydrogenotrophic methanogens) at pH close to 7. In response to a decrease in pH from 7 to 6.5, the acetate degraders were predicted to decrease in relative numbers and in absolute abundance, but to remain more abundant than the hydrogen utilizers. At pH ≤6, the acetate degraders were predicted to decrease to zero abundance. These observations are directly correlated with the given inhibition constant (pH value at which a 50% inhibition of growth rate is observed) of lower pH inhibition, which is six for acetoclastic methanogens but five for hydrogenotrophic methanogens [26]. Very small changes were predicted, in absolute abundance and relative abundance, of acidogens and acetogens at a decrease in pH from 7 to 6.5 (Figure 6). The relative abundances of sugar-degrading microorganisms in reactors fed CH-DFE and CH-1 were predicted to be on average 53.8 ± 2.5% and 59.5 ± 1.2%, respectively, at pH ≥ 6.5, and both increased with decreasing pH (Figure 6a,b). Other groups of microorganisms predicted showed no significant difference between reactors fed CH-DFE and CH-1, and no clear changes with pH (Figure 6).

## 4. Discussion

### 4.1. Overall Microbial Community Structure

The four UASB reactors studied here developed somewhat similar composition of the microbial community, with dominance of Firmicutes and Synergistetes and with lower relative abundance of additional phyla, such as Bacteroidetes, Chloroflexi, Proteobacteria, and Spirochaetes. Members of these six phyla are known to have quite broad metabolic capacities and to fulfill different metabolic steps (hydrolysis, acidogenesis, and acetogenesis) needed for a functional AD process [50,51]. These phyla are commonly found in studies using e.g., next-generation sequencing (NGS) on mesophilic UASB processes operating with different substrates, such as potato starch processing wastewater [52], sugar refinery wastewater [36], hydrolyzed spent grain [40], and trichloroethylene wastewater [53]. These phyla are also commonly found in different mesophilic continuous stirred tank reactor (CSTR) processes [50,54,55,56,57,58,59].

A typical feature of UASB processes appears also to be higher relative abundance of the phylum Euryarchaeota, which are normally found to be present in levels below 5% of the total community in CSTRs operating with different substrates, including when using the same primer sets as in the present study [36,44,52,60,61]. This higher share of archaea in UASB processes compared with CSTR processes might be explained by two essential differences between UASB and CSTR processes. Firstly, the formation of granular sludge in UASB processes, giving a longer microbial biomass retention time than in CSTR. Secondly, the use of soluble substrates. When enzymatic hydrolysis is used in a pre-step to solubilize cellulose and hemicellulose in lignocellulosic plant materials prior digestion in an UASB reactors, like in the current study, the need of hydrolytic microorganisms is lower compared to direct digestion of the plant material. Phyla Bacteroidetes and Firmicutes, which include many hydrolytic bacteria, had a lower relative abundance in the present study compared to the community in CSTR fed with lignocellulosic substrates [60]. The phylum Euryarchaeota contains several well-known methanogenic members that have been found previously in UASB, such as *Methanospirillum*, *Methanofollis*, and *Methanosarcina* [35], and species also found to be dominant in the present study, i.e., hydrogenotrophic methanogens belonging to the genus *Methanobacterium* and acetoclastic methanogens belonging to the genus *Methanosaeta* [50]. The low relative abundance of *Methanosaeta* compared with *Methanobacterium* is somewhat contradictory to findings in several previous studies of anaerobic granules that acetoclastic methanogens such as *Methanosaeta* and/or *Methanosarcina* are usually the core organism in anaerobic sludge granulation [62,63,64]. However, this confirms findings in other studies showing maintenance of granular structure without high abundance of these two genera [65,66].

### 4.2. Influence of Process Variables on the Microbial Community

During the experimental period, the reactors were exposed to changes in substrate and OLR, which caused both the performance and the microbiology in the different reactors to deviate somewhat. According to CCA analyses, the substrate itself was not the direct/main driving force for the development of the overall microbial community pattern. Instead, the concentration of organic acids, acetic and propionic acids, showed the most significant influences on the overall microbial community, followed by OLR, total COD fed, and undissociated propionic acid. During operation, all reactors at times showed the presence of acids above 0.5 g/L, but the number of occasions and the influence of these acids on the pH value varied between the reactors. While the occasional accumulation in 1A and 1B could be attributed to noted interruptions of recirculation, the acids accumulated more frequently in 2B and the accumulation could not be linked to clear interruptions to recirculation. Reactor 1A reached similar levels of accumulated acid as reactor 2B at the end of P4, but the alkalinity was not completely consumed and thus, acidification did not occur. The more stable pH observed in reactors 1A and 1B than in 2A and 2B was likely due to the higher alkalinity in CH-DFE compared with CH-1. As suggested in a previous publication, the difference in the overall composition of the substrates, with a lower sugar concentration in CH-DFE compared with CH-1, possibly also has an impact on acid levels [14]. However, this cannot be proven and needs further investigation. The degree of instability in reactors 2A and 2B was likely also influenced by the level of undissociated organic acids, which increased with decreasing pH. Undissociated organic acids can inhibit microbial cells by diffusing through cell membranes and decreasing the intracellular pH [67], and by other mechanisms [29]. Inhibition, which has been shown for propionate-oxidizing bacteria [32,33,34,35] and acetoclastic methanogens [68,69,70], can further reduce acid degradation rates and cause additional accumulation and inhibition.

The alkalinity in reactors fed CH-1 was apparently too low for the OLR applied in the present study. Although the alkalinity was higher in reactors fed CH-DFE than CH-1, it was still relatively low compared with that reported in other studies, where substrates with similar COD concentrations as in this study have been digested successfully in UASB reactors with either higher alkalinity in the reactor liquid or use of pH control [24,36]. In future studies with higher OLR than applied here, we recommend higher addition of alkali to both substrates, or a higher recycling rate, which can reduce the need for alkalinity [71].

Loss of microbial biomass, as noted for reactor 2B at acidification to pH 4.7, has also been observed in other studies upon acidification of UASB processes, with e.g., a 13% decrease in VSS in the UASB reactor after propionic acid accumulation to 1.9 g/L (pH not presented) [40]. The organic load per microbial biomass increases as microbial biomass is lost, at a constant volumetric OLR, compared with reactors with less biomass lost. Due to biomass loss in reactor 2B, the organic load to microbial biomass was roughly 45–89% higher than in the other reactors, at the same volumetric OLR, during P4. This is a probable reason for the higher acid accumulation compared with 2A.

#### 4.2.1. Correlation between Bacteria Groups and Organic Acids

According to the CCA and the relative abundance changes in bar plots, the phyla Synergistetes (represented by the genus *Aminivibrio*) and Proteobacteria (represented by the order Rhizobiales) were the dominant groups and showed a negative correlation to levels of organic acids, while levels of Thermotogae instead appeared to be positively correlated. In addition, the abundance of Synergistetes and Proteobacteria showed a distinct drop in reactor 2B at day 78, in line with the pH drop. It is difficult to know whether these observed correlations were the cause or consequence of acid accumulation. However, in line with the observed negative correlation with organic acids in the present study, a decreasing trend in these two phyla was also seen in response to increased OLR and organic acid level in a previous study on UASB processes, where the effect was independent of pH changes [36]. The observed decrease in abundance of Synergistetes and Proteobacteria with increasing acid level is also in line with findings in a study of degradation of macroalgae in a CSTR, where a decrease in relative abundance was seen when the acetic and propanoic acid level reached around 6 and 2.5 g/L, respectively, at neutral pH [72].

In the present study, the phylum Synergistetes was dominated by the genus *Aminivibrio*, while in previous studies of UASB processes, Synergistetes was dominated by unclassified Synergistaceae, *Thermanaerovibrio, Synergistes* sp., and *Aminiphilus*, respectively [36,40,52,53]. To our knowledge, the genus *Aminivibrio* has not been reported previously in UASB systems. One known member of *Aminivibrio* has been isolated from an anaerobic, propionate-oxidizing enrichment culture initiated with soil from a rice field [73]. This isolate of genus *Aminivibrio* is able to use amino acids and organic acids, but not carbohydrates, and to produce acetate, propionate, H_2_, and CO_2_ as major end products [73]. It grows at pH range 6.4–8.4 and growth can be enhanced in a defined medium when co-cultured with the hydrogen-utilizing methanogen genus *Methanobacterium* [73]. A recent stable isotope probing (SIP) experiment with 13-C acetate suggested that *Aminivibrio* can perform acetate oxidation in collaboration with hydrogenotrophic methanogens, including *Methanobacterium* [74]. Thus, the decrease in *Aminivibrio* could possibly indicate a decrease in acetic acid-degrading capacity caused by the inhibited *Aminivibrio* and/or decreased activity of the methanogen, negatively influenced by the organic acids and particularly, the undissociated acids. In the present study, hydrogenotrophic methanogens belonging to the order Methanobacteriales dominated throughout the study and were, thus, most likely important for driving the degradation of different organic acids, including acetic acid, all depending on low partial pressure of hydrogen [50]. The abundance of this methanogen was relatively stable over time across all reactor samples, but its activity could have varied over time and influenced acid degradation.

The order Rhizobiales dominated among Proteobacteria in the present study and was shown to be negatively correlated with the level of organic acids. Rhizobiales is an important denitrifying heterotrophic microorganism and is often found as a dominant group of bacteria in anaerobic granular sludge and biofilm in wastewater treatment plants [75,76,77,78]. A previous study proposed that one member within this group, *Kaistia granuli* sp. nov, has the ability to utilize acetate as a carbon source [75]. Little is known about the relationship between Rhizobiales and acetate and/or propionate level in AD processes. However, a more pronounced decrease in relative abundance of Rhizobiales, compared with *Aminivibrio*, was seen when the concentration of acids increased in the present study. Moreover, a specific unipolar polysaccharide adhesin was recently found to be produced by members of the order Rhizobiales and is believed to contribute to biofilm formation [79]. Thus, the observed decrease in this group might explain our finding that the granular bed in 2B was lower than that in the other reactors in P3.

At the end of experiment, besides the decreases in the abovementioned phyla, two uncultured clones, AUTHM297 (belonging to family Petrotogaceae, phylum Thermotogae), and NB1-n (belonging to class Mollicutes, phylum Tenericutes), appeared in reactors 2A and 2B, and were also positively correlated with the level of organic acids in the CCA. Clone AUTHM297 has been found in various AD systems, including UASB reactors [80], batch reactors [81,82], and anaerobic migrating blanket reactors (AMBR) [83]. There is no information available about the putative functions of AUTHM297, as no cultured representatives have been identified for this group. A recent study found AUTHM297 to be enriched at the end stage of a UASB process operating with a wood and hay mixture and selenium- and sulfate-containing synthetic mine-influenced water, while the organic acid concentration in the reactor was very low [80]. It has also been hypothesized that AUTHM297 has a specialized metabolism that allows it to survive in environments with highly degraded organic matter [80]. The relationship between AUTHM297 and methanogens has not been studied, but one study has pointed out that members of the Thermotogae can grow on acetate in the presence of the methanogen Methanobacteriaceae [84].

Clone NB1-n (phylum Tenericutes) has been found in aquatic sediments [85,86,87], rumen [85], a methanogenic AD process treating oil-containing municipal waste [88], and cellulose-rich substrate [89]. According to the recovered genome, there is no direct proof that NB1-n is involved in methane metabolic pathways, but it may have metabolic potential for fermenting simple sugars and producing small organic acids, such as acetic acids [85].

#### 4.2.2. Correlation between Bacteria Groups and OLR

The phyla Firmicutes and Bacteroidetes were found to be relatively stable over time, compared with other phyla, across the four reactors. According to CCA, these two phyla were more positively correlated with the increase in OLR than the level of organic acids. As mentioned in Section 4.1, members of these two phyla are known to be involved in several of the essential steps needed for AD. Thus, they typically remain dominant even during process turbulence [56,59,90,91]. However, at day 203, an unclassified genus belonging to the family Eubacteriaceae within the phylum Firmicutes was found to be enriched in reactor 1A, while the genus *Alistipes* within the phylum Bacteroidetes was found to be enriched in 2A.

The family Eubacteriaceae has been detected in previous studies of UASBs receiving food processing wastes and CSTRs operating with straw and cow manure as substrates, but in very low relative abundance (<1% of the total bacterial community) [92,93]. In a study of UASB processes operating with molasses wastewater, Eubacteriaceae was found to be one of the dominant phyla in the total bacterial community, and its relative abundance increased from 19.1 ± 6.2% to 30.5 ± 8.7% when OLR was raised from 3.6 to 5.5 COD g/L/d, but started to decrease again with continued increase in OLR [94]. Eubacteriaceae contains many acetogenic members that tolerate rather high content of acetic and propionic acids, and growth of members of this family has also been shown to be favored by H_2_ supplementation [68,69,70,95,96]. Thus, one possible hypothesis could be that high OLR gives a high flow of substrate (and H_2_), which in a certain range is beneficial for members of this family.

Members of *Alistipes* (phylum Bacteroidetes) have been found previously in very low relative abundance in CSTRs co-digesting fish waste and cow manure, and in a UASB treating synthetic wastewater containing trichloroethylene (TCE) [53,97]. In the latter case, the relative abundance of *Alistipes* stayed at around 1% despite a gradual increase in OLR (from 2.9 to 14.4 COD g/L/d) [53]. Some members of *Alistipes* can also perform acetogenesis and can produce succinate, acetate, and propionate as the main fermentation products [98,99,100]. However, based on known information and the process data in the present study, it is still unclear why *Alistipes* was particularly enriched in 2A.

#### 4.2.3. Correlation between Bacteria Groups and Total COD Fed

In addition to OLR, the term “total COD fed” (Appendix A) was introduced as a variable in the CCA. The reasoning behind this was that microbial growth is related to the total amount of metabolized substrate, which can be particularly relevant in the analysis of microbial populations in UASB processes where part of the biomass, active or not, is retained in the process. Moreover, as the OLR was varying over time, with some periods even without feeding, we wanted to investigate if total COD fed would have a stronger correlation to some microbial groups as compared to instantaneous OLR. A correlation between a microorganism or group of microorganisms with total COD could show a collective response to the fed given, since the start of the experiment, e.g., both load and change in substrate composition. With this approach, we identified the phylum Planctomycetes and clone vadinHA49 to be valuable to study in more detail. This clone and phylum were strongly correlated with total COD fed and increased significantly in reactors 1A and 1B at the end of the experiment. The vadinHA49 clone was first found in a fluidized bed anaerobic digester fed with vinasses [101], and later, in the intestinal tract of insects [102,103], in an acidic boreal bog [104], and in sediments in a sulfur spring [105]. Studies on vadinHA49 are fairly limited and its putative metabolic functions remain unclear. Planctomycetes were also found at the end of a previous 218-day study (it was not reported whether it was present earlier) of the microbial community of UASB reactors using a substrate with rather similar COD concentration of the substrate and OLR as in the present study: COD 20 g/L (compared with 34–36 g/L in the present study) and an OLR of COD 13 g/L/d (compared with 8–9 g/L/d in the present study) [52]. Two other phyla had also a strong correlation to total COD fed. Those could also be valuable to study further, despite low abundances below 1%.

#### 4.2.4. Correlations between Process Variables and Archaea Groups

At the genus level among the Euryarchaeota, the increase in relative abundance of *Methanobacterium* after acidification of 2B in P2 confirms previous findings of relatively high tolerance of a member of *Methanobacterium* (*M. formicicum*) to acids [37]. In that study, the concentration 0.17 g/L of undissociated propionic acid caused 50% inhibition of methanogenic activity. Exposure to 0.69 g/L undissociated propionic acid caused more severe, but still reversible, inhibition [37]. However, even though the relative abundance of cells of *Methanobacterium* in the present study appeared to increase after acidification, the qPCR showed stable levels. Thus, the observed increase in abundance might have rather been an effect of a comparably lower washout i.e., the relative abundance of *Methanobacterium* increased due to decreased abundance of other methanogens. Regardless, this result does not imply that the activity of the cells was maintained or that they were even alive. The results in [37] also suggest that *Methanobacterium formicicum* could have become irreversibly inhibited after exposure to 0.86 g/L of undissociated propionic acid. The genus *Methanosaeta* is an obligate acetoclastic methanogen, and it is more sensitive to pH drops and acetate accumulation than *Methanobacterium* [106,107]. This might explain the decrease in *Methanosaeta* in 2A and in 2B after acetate accumulation and pH drops below 6 (Appendix A). As *Methanosaeta* was the important acetate-utilizing methanogen in the AD processes studied, its decline could also have intensified acetate accumulation in the present study.

### 4.3. Possible Bias Introduced by Using Two Different Primers for Archaeal Community

Two different primer sets (515’F and 805R, 516F and 915R) were employed to reveal the overall microbial and archaeal community, respectively, in the present study. These two primer sets have previously been shown to adapt well to Illumina sequencing. Moreover, the archaeal community generated from these two primer sets is generally consistent, but with higher resolution in the results for primer set 516F and 915R [44,61,108,109]. However, the results of these two primer sets were somewhat contradictory for sample 1A_204. With the universal primer set, the relative abundance of archaea in 1A largely decreased from day 153 to 204 (from 25.2 to 1.2%) (Figure 5), despite the opposing trend found for archaeal richness in 1A_204 (Appendix A). This could have been caused by differences in the design of these two primer sets, as primer set 516F and 915R is designed to specifically target the archaea community, while primer set 515’F and 805R targets both bacteria and archaea and could miss some archaeal species [42,43,110,111]. Furthermore, archaeal primer set 516F and 915R produces longer amplicons (around 400bp) than universal primer set 515’F and 805R (around 300bp), providing more information for blasting in the database and thus, in theory, helping to distinguish some rare species [112].

In the present study, these missing species for the universal primer set seemed to belong to the phyla Bathyarchaeota and Woesearchaeota (Figure 5 and Appendix A). The roles of these two phyla in the AD process are yet to be determined, but a recent study found some members of the nonmethanogenic phylum Bathyarchaeota carrying a gene encoding acetate kinase, indicating that it can grow as an acetogen [113]. Other studies indicate a syntrophic association between these two phyla and acetoclastic methanogens [114,115].

The phylum Woesearchaeota has been identified in surface waters, sediments, and methane-containing Arctic permafrost [116,117,118], and in wastewater treatment plants [119]. Complete genomes of the representatives reveal that Woesearchaeota is capable of transferring acetyl-CoA to acetate for ATP production using acetate kinase and phosphate acetyltransferase [118], indicating its ability to utilize acetate. Representatives of Woesearchaeota lack genes encoding enzymes needed for many critical metabolic pathways, but possess genes for precursors involved in pathogenesis in bacteria, indicating these archaea might have a symbiotic or parasitic lifestyle [117,118]. In an AD process, the presence of these two phyla might provide extra metabolic pathways linking the use of acetate and methane formation.

Besides the above two phyla, archaeal phylum WSA2, represented by class WCHA1-57, increased at the end of the experiment across all reactor samples except 2B, where it still represented less than 1% (data not shown). WCHA1-57 is suggested to play an important role in the conversion of propionate to methane and has been found to be enriched in cultures fed formate or H_2_/CO_2_ [120,121]. In our case, lack of this archaeon might have contributed to the accumulation of propionate observed in reactor 2B.

### 4.4. Relations between ADM1 Predictions and Experimental Data

Using Monod kinetics and known inhibition parameters from the frequently used ADM1 model [26], the effect of pH on metabolite production and population dynamics was plotted (Figure 6) and compared against the fermentation data (Figure 1) and microbial community data (Figure 5), with and without dark fermentation pretreatment. The theory of organic acid inhibition [30], as simulated using a standard pH inhibition equation in the ADM1 model, showed increased acetate and propionate levels and a shift in relative abundance in the microbial community.

The dominant archaea found in the present study, *Methanobacterium* and *Methanosaeta*, have been well studied and are known as a typical hydrogenotrophic methanogen and an acetoclastic methanogen, respectively [122,123]. The ADM1 model predicted a drastic decrease in the relative abundance of acetate degraders with a decrease in pH to below 6.5 (Figure 6), which aligned with the measurements of decreased *Methanosaeta*. In contrast to the measured results on acetoclastic and hydrogenotrophic methanogens, the ADM1 model predicted higher abundance of acetate degraders compared with hydrogen degraders at pH above approximately 6.3. The modeling results agreed with previous findings on microbial composition in granules, which suggest that acetoclastic methanogens should be more dominant when a UASB is operated around neutral pH [63,64]. Moreover, since the division rate is relatively low for microorganisms with anaerobic metabolism, and it could, therefore, take a long time for the microbial communities to develop, it is unknown whether the archaeal community would develop towards that predicted by ADM1 if the process were operated for a longer time at the given OLR (COD 6 g/L/d) or whether the model needs development to reflect the reality.

The specific relative abundance of bacteria in the model and in the measured data was not compared, since the microbial study was based on 16S rRNA genes and one bacterial phylum can cover more than one functional group. To evaluate the dynamics of functional groups for bacteria, in-depth activity studies on the level of e.g., transcription and protein expression, are needed.

## 5. Conclusions

Inoculum from a brewery wastewater treatment’s internal circulation reactor adapted well for the degradation of combined hydrolyzed lucerne and effluent from dark fermentation of hydrolyzed wheat straw at an OLR of COD 8.5 g/L/d. The alkalinity of the hydrolyzed wheat straw and lucerne used as substrate in two reactors was too low for the applied conditions. Severe process acidification (to pH 4.7) in one of them caused loss of microbial biomass and strongly influenced microbial community composition, even at 47 days of operation after an acidification event. Hydrogenotrophic methanogens of the genus *Methanobacterium* were confirmed to be relatively resistant to acidification and increased in relative abundance, while many bacterial phyla and the archaeal genus *Methanosaeta* decreased in relative abundance at acidification. Some little-known phyla increased at the end of the operating period. Future studies should determine their development over longer periods and explore their function, as well as that of many of the other microorganisms detected in this study. It would also be valuable to study how microbial communities develop under longer periods of stable operation with the substrates used in the present study. The changing environment over time in the current study makes it difficult to make any conclusions in this direction. The ADM1 model results were in line with the observed decrease in the genus *Methanosaeta* when the pH decreased below 6.5. However, more in-depth studies and parameter optimization are required to increase the precision of the model.

## Figures and Tables

**Figure 1 microorganisms-08-01394-f001:**
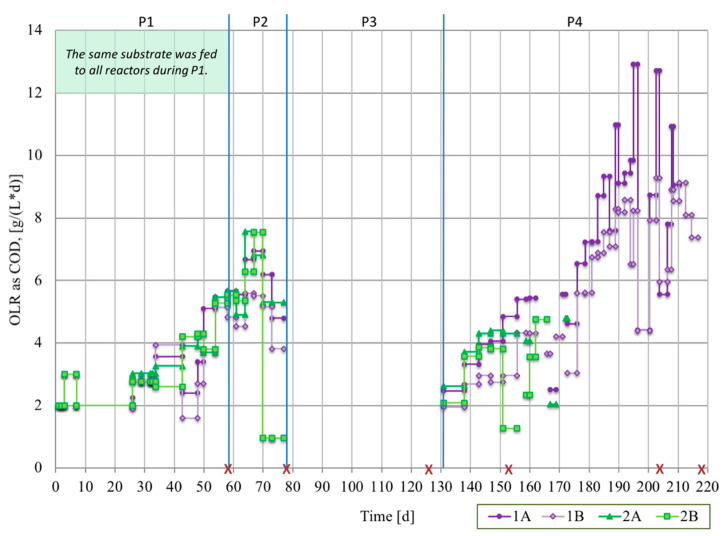
Organic loading rate (OLR) of upflow anaerobic sludge blanket (UASB) reactors 1A, 1B, 2A, and 2B, expressed as chemical oxygen demand (COD), per reactor volume and time (g/L/d). Average OLR (horizontal line) between time points (dots). The blue lines indicate the changes between operating periods 1–4 (P1–P4), described in Table 1. Time points of microbial sampling are marked with red crosses.

**Figure 2 microorganisms-08-01394-f002:**
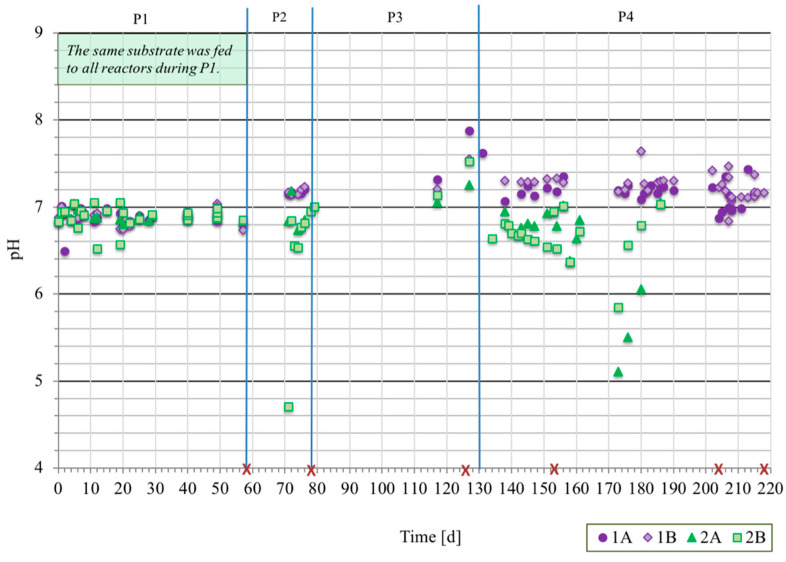
Values of pH recorded in liquids of upflow anaerobic sludge (UASB) reactors 1A, 1B, 2A, and 2B. Time points of microbial sampling are marked with red crosses. The blue lines indicate the changes between operating periods 1 to 4 (P1–P4) described in Table 1. Abbreviations: d—days.

**Figure 3 microorganisms-08-01394-f003:**
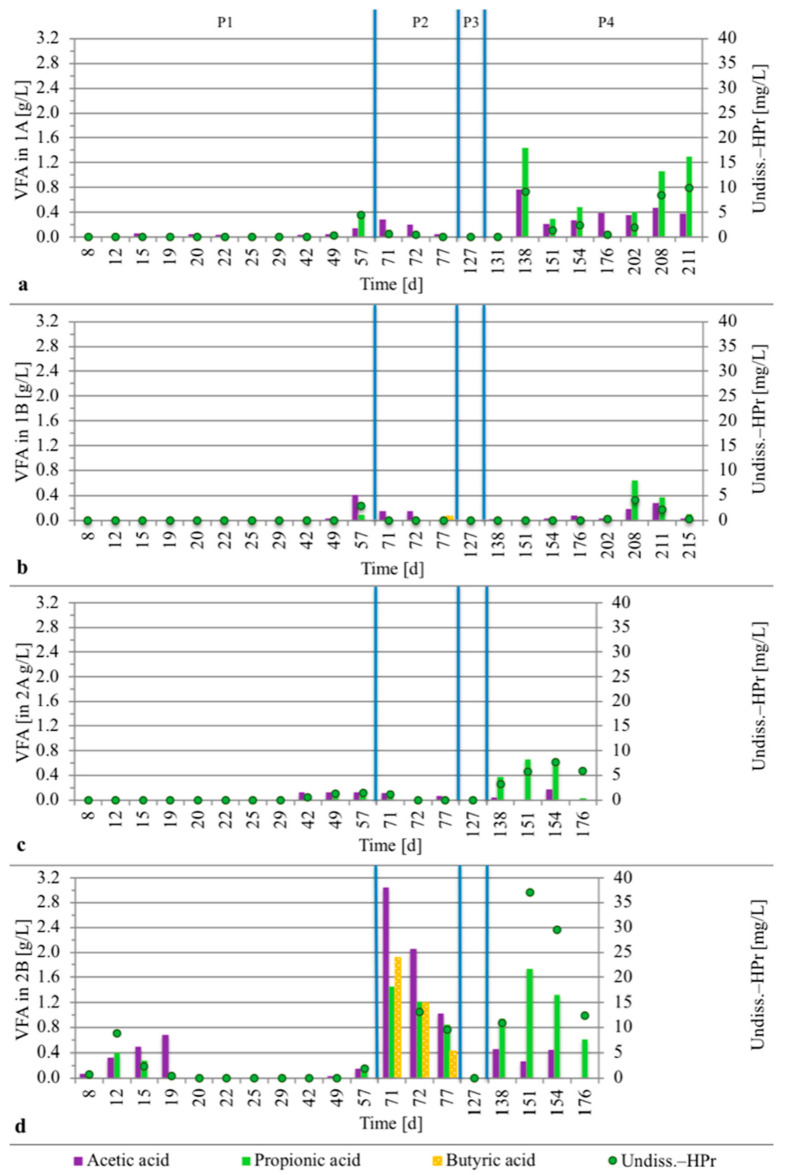
Concentrations of acetic, propionic, and butyric acid (left y-axis) and undissociated propionic acid (right y-axis) in liquids of upflow anaerobic sludge (UASB) reactors: (**a**) 1A; (**b**) 1B; (**c**) 2A; (**d**) 2B. In reactor 2B, the concentration of undissociated propionic acid on day 71 was off the scale (864 mg/L). The blue lines indicate the changes between operating periods 1 to 4 (P1–P4), described in Table 1. Abbreviations: Undiss.-HPr—undissociated propionic acid; VFA—volatile fatty acid; d—days.

**Figure 4 microorganisms-08-01394-f004:**
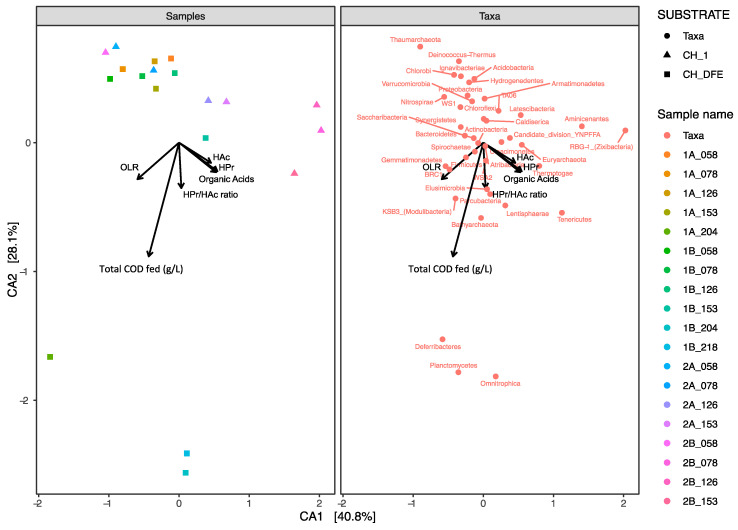
Canonical correspondence analysis (CCA) of amplicon sequence variants (ASVs) at phylum level of the 16S rRNA genes in upflow anaerobic sludge blanket (UASB) reactors 1A and 1B after 58, 78, 126, 153, and 204 days of operation, and in reactors 2A and 2B after 58, 78, 126, and 153 days of operation.

**Figure 5 microorganisms-08-01394-f005:**
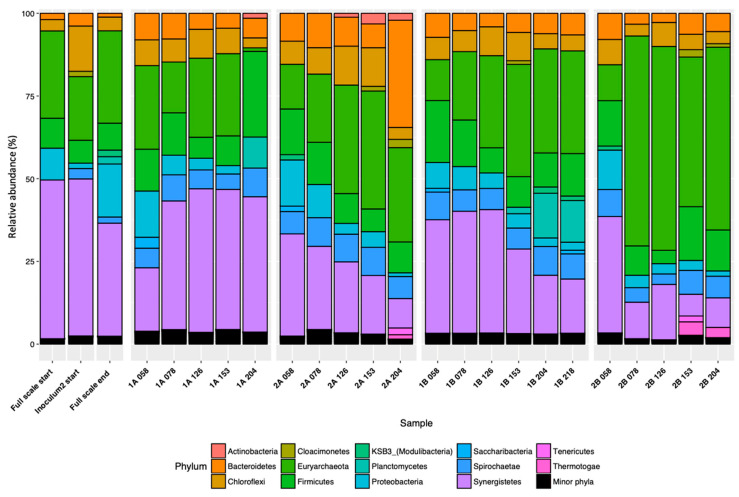
Relative abundance of microbial 16S rRNA genes at phylum level in upflow anaerobic sludge blanket (UASB) reactors 1A, 1B, 2A, and 2B, arranged by operating day (day 58, 78, 126, 153, 204, and additionally, day 218 for 1B) and the inoculum used for reactor set-up (two types of granules, labeled ‘Full scale start’ and ‘Inoculum2 start’), and granules from the same full-scale plant when our experiment ended, labeled ‘Full scale end’. Phyla present in relative abundance less than 1% are merged into ‘Minor phyla’ (The data presented in Figure 5 are also visualized as an abundance heatmap in Appendix A).

**Figure 6 microorganisms-08-01394-f006:**
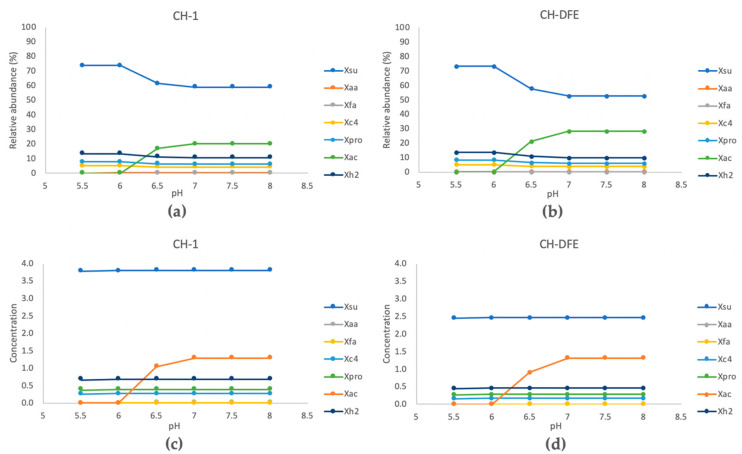
Key microbial population dynamics predicted by anaerobic digestion model no. 1 (ADM1), divided into four functional groups: acidogens (Xsu, Xaa, and Xfa), acetogens (Xc4 and Xpro) hydrogen degraders (Xh2), and acetate degraders (Xac). (**a**), (**b**) Predicted relative abundance and (**c**), (**d**) predicted absolute abundance.

**Table 1 microorganisms-08-01394-t001:** Description of operating periods of upflow anaerobic sludge blanket (UASB) reactors.

Period	Day, Start-End	Description
P1	1–57	Substrate CH-1 to all reactors. OLR gradually increased (Figure 1). Temperature increased from 32 to 37 °C over the first 5 days.
P2	58–77	1A and 1B switched to substrate CH-DFE on day 58 and continued for the whole experiment. 2A and 2B continued with substrate CH-1. Average OLR of COD 5.4 g/L/d (SD 0.5) during days 59–77 for 1A and 1B, and days 58–67 for 2A and 2B.
P3	78–130	Starvation. OLR = 0 for all reactors. Temperature reduced stepwise to around 25 °C in the beginning of P3 and increased at the end of P3 to: 31 °C on day 118, 35 °C on day 125, and 37 °C on day 130. Bed height determined on day 117.
P4	131–217	The OLR was started at COD 2.3 g/L/d (SD 0.3) and thereafter changed, as illustrated in Figure 1. pH of substrate CH-1 adjusted to 7.1 (instead of 7.0) on day 143–160 and to pH 8.3 on day 161–177. Feeding terminated for 1A on day 211, 1B on day 217, 2A on day 173, and 2B on day 168. Average OLR 8.5 (SD 0.2) during days 183–201 for 1A and days 208–217 for 1B.

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
