# Peer review of "Diversity and Abundance of Microbial Communities in UASB Reactors during Methane Production from Hydrolyzed Wheat Straw and Lucerne"

_microorganisms, 2020, doi:10.3390/microorganisms8091394_

Round 1
Reviewer 1 Report
Microorganisms 88127: Development of microbial communities during methane production of hydrolyzed wheat straw and lucerne in UASB reactors
Abstract
P26: What do you mean by “accumulated COD fed”. I can only imagine that this refers to the amount of COD that have been fed to the reactor at a particular time from day 0. However, this correlation does not much make sense to me.
Introduction
Overall statement: The introduction is interesting and well-written. However, it is totally disconnected to the paper main topic “UASB microbial community”. If you decided to split the paper in two, this implies that the introduction (as the paper) should mainly focus on microbial community and AD microbial community response to acidification events. Similarly, nothing is said about AD/UASB process modelling. Accordingly, the introduction needs to be improved.
L56-59: References are needed. Utilise the knowledge from the previous publications to enrich this introduction.
L74: VFA pka is 4.8, so the dissociated form would prevail (> 98%) until pH 6.0. It is best to give numbers that subjective term like “neutral pH”
Materials and methods
L105: Regarding CH-1, CH-DFE and other feedstocks, where they continuously produced or produced in a single batch before the experiment. If the second option, how was it stored? This is important because bugs immigration may have a role in the reactors microbial community. I am thinking about “accumulated COD fed”
L157: State if this was intentional or unintentional “on a few occasions feeding continued despite a recorded pH 6.6–6.7,”
Table 1, P3: I am still wondering if this was done on purpose. There is no problem since problems always happens when you run reactors for over 200 days. Even more, there are a couple of publication showing than UASB granules keep activity over periods with no feeding.
Table 1, P4: increased from 0 to…
Figure 1: I was expecting more sampling events. Note that with just 6 sampling events the conclusions should be carefully done
L183: Thank you for the honesty “was off (by mistake)”
L196: catheter tip syringe?
L203: “methane production” because AMPTS does not measure methane composition
L241: Was the ADM1 modified to adjust to a UASB? As you are well-aware UASB SRT is much higher than a CSTR SRT and consequently the microbial community dynamics. Please, explain in the manuscript how this was implemented. Including other modification of the default ADM1.
Results
L285: lost through/with/in the effluent
L292: check significant figures, decimals not really needed
Figure 3: Could be moved to supplementary material. Particularly if repeats data in [1]
- It would be nice to include a scheme of the UASB set-up in the supplementary material
L308: Indicate pH value “reactor was acidified on day 71 (pH of …)”
L300: Period 4 was quite a period. No action needed, but How much time of stable operational time do you think it is needed to have a representative granule microbial community? You sampled on day 204 and 218, however, since reactor parameters changed quite a bit, I am wondering how representative are the microbial community results.
L337: Why did you split bacteria and archaea? What would happen if you do the analysis on the entire microbial community? It may be worth to include a sentence or two at the end it is a synergistic microbial community.
Figure 4: I am still surprised about this “accumulated fed COD”. Is this truly a factor? I guess correlation exist because this parameter has the highest variability of the tested parameters. Similar thing would happen if you include time as a variable? Please, try! But, is time a true variable? Correlation does not mean causation, right?
Figure 4: A bit surprised that this figure was done at phyla level because of the poor bug-function resolution at such high taxonomic level. Could you at least include one done at genus/specie/ASV level in the supplementary material?
Figure 4: BTW, how did you normalise/standarise the weight each factor? Meaning, to make them factors even so one does not have a larger impact than others.
Figure 5: Heatmap please! It hard to compare and extract data from these bar graphs. Please, convert it to a heatmap. There are several R packages to do so.
Figure 5: Could you include a heatmap at ASV level in the supplementary material as well?
L377: Nearly all methanogens in AD are Euryachaeota so this sentence does not privde much information. Please, zoom in (genus level?), at least for methanogens
L453: There is no VFA concentration inhibition function in the basic ADM1. Did your model include one? If so, describe in the materials and methods.
L453-461: It may be worth stating here or in the materials and methods which pH inhibition function was used for methanogens and bacteria as well as their pH_upper and pH_lower inhibition values
Discussion
L471-472: This is true for nearly any AD. Not only UASB.
L480: Have you considered the more soluble nature of UASB feedstocks?
L489-493: What is the archaea profile for acidified digesters in the literature? It may be worth to compare them with your results
L502: consider my previous comment about “accumulated COD fed”
L502: Did you include pH as variable? I cannot recall
L628: Consider re-wording or deletion according to previous discussion.
L636: This still does not make sense to me. Could you provide a hypothesis to explain it? Or similar observation from other research field e.g. soil, human gut. (“Thus, it appears that this phylum benefits from a total high load of organic matter.”)
L649-L652: This is a big limitation of 16s sequencing, particularly when using high SRT systems. However, imply that your results are useless (I do not think so!). Have you considered that the relative abundance of Methanobacterium increased because other bugs abundance decreased? Meaning Methanobacterium is less sensitive to pH changes than other bugs so, it the end, its relative abundance increases. Just a hypothesis.
L683-703: Nice discussion but this does not fit the Section title. Note that the manuscript is quite large already.
L722: Division time is not that long for not been captured in an the model not after 200+ days of operation.
Conclusions
Modify according to previous discussion. Also, little is said about the microbial community which was the main topic of this publication.
Reviewer 2 Report
Review comments on “Development of microbial communities during methane production of hydrolyzed wheat straw and lucerne in UASB reactors”
General comment:
To me, a slight change in the title “The diversity and abundance of microbial communities in UASB reactors during methane production from hydrolyzed wheat straw and lucerne” explains the content of the manuscript better.
Specific comments:
Typographical error: Be consistent in using 16S rRNA and not “16s rRNA” throughout the manuscript
Figure legend for S1: Check for the appropriateness of this sentence “Reactor name and time, in days after start of experiment and in minutes (m) after last feeding period, are indicated on the x-axis.”
Figure legend for S3: Check for the appropriateness of this sentence “The blue lines indicate operating periods 1 to 4 (P1-P4), described in Table 1.”
Figure legend for S4: Write the full form of HAc. Eq./PA in the figure legend.
Tables S1 & S2: Provide a short table legend withing the worksheet and also mention what are 0D and 1D.
Figure6: Provide a high-quality figure
Provide all primers information as supplementary table along with size of the PCR products etc.
Line32: Keywords: Change “biogas production” to “methane production”
Line91: Be specific here instead of “with plant hydrolysates”
Lines212-217: Mention in a sentence or two about what happens after the PCR amplification, such as, was there a gel run for size confirmation and excising the bands for purification or the PCR product was purified directly by other methods, and then the sequencing reaction was performed, if so, how many primers were used for sequencing etc.
Line551: Typographical error: “Synergistes sp.” should be “Synergistes sp.”
Round 2
Reviewer 1 Report
881217_v2: Diversity and abundance of microbial communities in UASB reactors during methane production from hydrolyzed wheat straw and lucerne
Introduction
Overall statement: My main concern about the introduction has not been addressed. In L103 you say “The aims of the present study were to obtain novel information on microbial communities in UASB reactors…” and “he aims further included understanding biology in acidification events”. However, nothing is said in the introduction regarding this topic. Which microbial communities have been found in granules? How substrate, load, up flow velocity have affected microbial composition, structure, function? How granules have responded to acidification events? Introduction is meant to give the reader the minimum background information needed to understand the manuscript, which this introduction in its current form still fails to do. L47-57 goes in this direction, but I find it too generalist and not targeted to UASB.
L36: use of wheat straw “residues” for
L47-57: A bit weird that you make all these claims from one auto-reference.
L115: OLR? according to L25
Materials and methods
L124: “new” or “additional” instead of “novel”
L137: 0.12 g/L
L149: subindex for “2” in ZnCl_2 and CuCl_2
Table 1: SD should go after units > 2.3 g/L/d (SD 0.3). It looks a bit odd in its current form
Figure 1: Improve the quality of Figure 1, it is difficult to differentiate between series
L286: “implicit in the lower pH inhibition expression”
Results
L303: remove “analysed" since you analysed the main ones.
Figure 2: Same quality issue as in Figure 1 above
Figure 3: try to improve this figure, 1) what means the black line surrounding the blue line?, 2) why there are not horizontal lines in Fig. 3d, 3) why in Fig 3b there the green dot is surrounded by a black dot, 4) and so on
Figure 5: The problem with bar chars is that it is impossible to compare different samples since in each sample each bug starts in a different height. On the other hand, in heat map each box is independent from each other while colour intensity goes from 0 to 100. Heatmap also allows including multiple R tricks. That is why I prefer (and encourage) heatmaps. Let’s leave it as it is.
Figure 5: Y-axis label should be “relative abundance (%)”
Discussion
L680: The concerns regarding the suitability of using “total COD fed” as a variable remain. At least, the explanation given in L681-702 helps to understand your vision. The problem is “total COD fed” only increases because of operation time.
L714: “This result does not imply that the activity…”
Round 3
Reviewer 1 Report
Good work! Ready to be published